# Critical Performance Aspects of Retrofitting Apartment Buildings Using a Multifunctional Façade System

**Satu Paiho \*, Tuomo Ojanen, Isabel Pinto Seppä and Martta Paavola**

VTT Technical Research Centre of Finland Ltd., P.O. Box 1000, 02044 VTT, Finland

\* Correspondence: Satu.Paiho@vtt.fi; Tel.: +358-50-331-5160

**Abstract:** There is huge market potential for energy refurbishment solutions in European buildings. This paper analyzes the challenges related to using a multifunctional energy efficient façade system, the "Meefs" system, in retrofitting multi-family apartment buildings. Similar challenges often occur also in other comparable façade renovation solutions. The focus is on hygrothermal performance even as other aspects are also discussed. After introducing the hygrothermal performance challenges of the Meefs system, numerical case analyses are performed in three different climatic conditions. The results for 26 cases are presented. A core result is that the drying of any exterior renovation system is mandatory to guarantee safe operation in different European climate conditions. This emphasizes proper design needs in all climates. Based on the analyses, design recommendations can be given for the Meefs system. In Central and Northern European climates, the system contains challenges which may hinder implementation in practice.

**Keywords:** façade renovation; residential buildings; hygrothermal performance; performance aspects

---

## 1. Introduction

There is a huge need to renovate European buildings [1]. This is supported by the number of past and on-going research and development (R&D) activities in Europe [2]. The renovation will support both improving people's living conditions and achieving wider climate targets; currently, buildings account for 40% of total primary energy consumption and for 36% of greenhouse gas emissions [3]. Furthermore, an ageing population, migration, and changing lifestyles and family structures all place new demands on housing [4]. New construction in Europe represents about 1% of the building stock annually [3], while the renovation rate has been quite low at 1.2% per year [5], resulting in initiatives to boost it [6]. Even if many barriers are non-technical [7] and recent projects have developed new solutions [2], reliable and easy-to-install technologies are still needed.

Various technologies, solutions and measures exist for energy efficient renovation of buildings, e.g., Xing et al. [8]. Many of these deal with building envelopes, e.g., Sadineni et al. [9]. Lewandowski and Lewandowska-Iwaniak [10] describe the optimal thermal and optical properties of external walls of a passive building. Unfortunately, simultaneous fulfilment of these requirements is difficult and in practice sometimes impossible.

Hradil et al. [11] analyze the durability of refurbished outdoor walls on the basis of building-physical analyses and estimate the benefits of refurbishment on concrete façade in Nordic conditions when an exterior insulation layer is added. The major part of building envelope failures was caused by excessive moisture content of building materials. This infers that moisture performance is one of the crucial aspects in envelope renovation, at least in cold climates.

External thermal insulation composite systems (ETICS) are a set of construction elements consisting of certain prefabricated components being applied directly to the façade [12]. They are utilized widely

in façade renovations in many countries [13,14]. The system is not modular. Basically, it includes only an extra insulation layer, thus improving the façade U-value, but without any other issues improving the building energy performance.

Utilizing prefabricated façade elements can provide many benefits in building renovations. Heikkinen et al. [15] define basic principles for the energetic modernization of the building envelope using prefabricated large-sized timber frame elements (the Timber-based Element System or TES method). There the basis for the use of prefabricated retrofit building elements is a frictionless digital workflow from survey, planning, off-site production and mounting on site based on a precise initial 3D measurement. Cronhjort [16] presents a pilot project where the prefabricated TES Energy Facades were utilized for the first time in Finland. Cronhjort and le Roux [17] present the second building in Finland in which the façades have been retrofitted using timber-based elements, the TES-system. The cladding and often also the existing thermal insulation are torn down when installing the TES-system. This slows down the renovation process even if the TES elements themselves are prefabricated.

Friege and Chappin [18] indicate that the (socio-economic) energy saving potential and profitability of energy efficient renovation measures is lower than generally expected. However, renovating buildings offers a good opportunity to consider improving energy efficiency. Systemic retrofitting can be applied in buildings and affordably maintained with minimum disturbances to end-users [19]. Systems and solutions specially designed for systemic retrofitting reduce the end-user inconveniences even further.

Façade renovation is often quite expensive [20,21]. When analyzing costs of holistic energy efficient building renovations in a Moscow residential district, Paiho et al. [22] show that façade-related renovation costs formed a major share of the total costs. Mata et al. [23] highlight that energy price developments have lower impacts than interest rates on the techno-economical potentials of different energy-saving measures. Still, means and solutions reducing the initial investments could support wider realization of these renovations.

In addition to energy efficiency improvements, energy renovations could provide other benefits, such as improved indoor environmental quality [24,25]. Results from Finland show that if applied on a national level in the building stock, energy-efficiency improvements would also have a positive effect on GPD and employment in the medium to long term [26].

The need for renovation and modernization of housing properties in Finland mainly concerns apartment buildings built in 1960s and 1970s [27]. Around 70% of the Finnish apartment buildings built during 1965–1995 need some kind of a façade renovation, although light renovation methods can be applied in 6–22% of the buildings [28]. In Germany, the progress of upgrading external walls is much slower than upgrading windows or roofs [29]: insulation measures have been carried out for about 20% of the wall area, the annual rate is below 1%, but the average insulation thickness has increased from 8 cm to 14 cm. In Spain, multi-family residential buildings, built between 1940 and 1980 and comprising about 50% of the housing stock, represent the greatest potential for energy savings due to the lack of technical standards in the energy efficiency field in the construction period and the low investment in conservation and maintenance carried out during their service life [30].

The above shows that various technologies, including façade solutions, exist for energy efficient building renovation. In addition, building physics issues are often crucial for their implementation. The review also shows that there is room for more advanced façade renovation solutions, but their performance issues need to be carefully evaluated. In addition, there are many apartment buildings in Finland, Germany and Spain that are in need of façade renovations. Even though the building physics issues and ensuring the safe moisture performance of the renovation system should always be studied, many projects aiming at energy improvements tend to leave these issues aside without relevant evaluation.

This paper evaluates performance aspects of a multifunctional energy efficient façade system (Meefs). The focus is on addressing the critical performance issues, which need careful design. The main goal is the evaluation of the hygrothermal performance of the retrofitting system through numerical simulations. The aim is to study the requirements that the safe moisture performance sets for the

façade system applied under different European climate conditions. The system has to be able to dry out both the moisture load from indoor air transferred through the renovated wall and the additional moisture accumulated in the old structure before the installation of the Meefs system. Depending on the climate loads, the initial moisture of the old structure may represent the highest load that has be taken into account in the design of the system. The exterior climate conditions define the needed drying efficiency of the system.

The remaining sections of this paper are structured as follows. Section 2 briefly presents the multifunctional energy efficient façade system for building retrofitting (the Meefs system), evaluates the critical issues of the Meefs systems, and then focuses on the target apartment building stocks in Finland, Germany and Spain. Section 3 discusses hygrothermal performance challenges of the Meefs retrofitting system. Section 4 analyzes the hygrothermal performance of the Meefs system in three different European climates. Section 5 deals with discussion and conclusions.

## 2. The Meefs System and Target Buildings

Firstly, this section presents the new façade renovation system which is analyzed in the remaining sections. Secondly, the critical performance aspects of the Meefs system are briefly evaluated. Later on, the hygrothermal performance is analyzed more in detail. Thirdly, the target building stocks are described in three European countries.

### 2.1. The Multifunctional Energy Efficient Façade System for Building Retrofitting - The Meefs System

Figure 1 shows the core idea of the Meefs system, i.e., it is an energy efficient non-intrusive façade concept based on modular technology components that will allow integrating both active and passive technologies in the façade without tearing down the exterior parts of the existing structure. Every module will represent different energy efficient innovative solutions and will be packaged into easy-to-install panels [31].

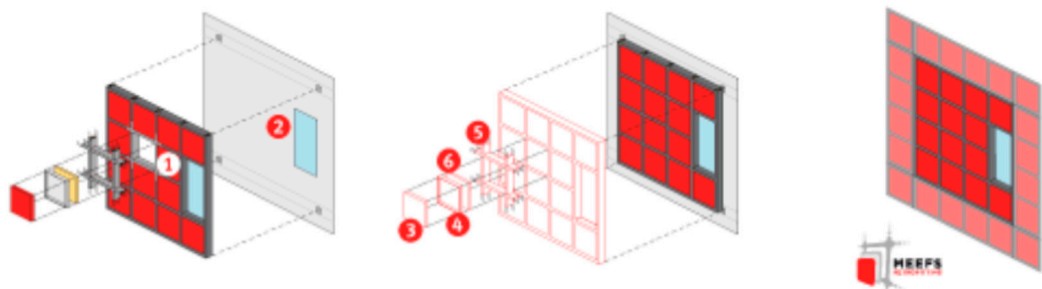

**Figure 1.** Meefs Façade System Constructive Process. 1) Multifunctional panel, 2) Existing façade 3) Technological unit 4) Structural module 5) Structural panel 6) Thermal insulation 2+4) Technological module 3+4+5) Multifunctional panel (The figure is retrieved from the Meefs project website [32].).

The system relies on industrialized production always using standardized panels, easily assembled technological modules, still allowing for personalized configurations for each façade typology, orientation and local climate conditions.

The façade system is composed of:

- **The structure** (Figure 1, 4+5): the Meefs system structure is based on lightweight and cost-effective structural panels and an anchorage system for fixing these structural panels to the existing façade.
- **Operating Control System**: the façade integrates an intelligent control system for the energy management and control of all the mobile elements. This is integrated in the building energy system.
- **Technological modules** consisting of structural module and technological units (Figure 1, 3+4+5). All the modules integrated in the façade include a particular technology or flexible combinations of technologies allowing the reduction of primary energy, either by reducing the energy demand of the building or by supplying energy by means of renewable energy sources (RES).

- **Back layer insulation**: this insulation layer is fixed at the back of each multifunctional panel.

The technological units in the Meefs system cover the main functions of a traditional building envelope and also add new functionalities as energy production and its flows management. Calculations by Paiho et al. [33] showed that in a realistic case the Meefs system could reduce the total heating energy demand of a typical Finnish apartment building from the 1970s by almost 40% annually and the space heating consumption by 57%.

### 2.2. Critical Issues of the Meefs Systems

In Table 1, thermal and moisture properties and in Table 2 other properties of the Meefs system are evaluated, in reference to the relevant scientific literature. Altogether six property classes are considered, namely thermal, structural, fire, moisture, cleaning, and data security properties. The tables present a general performance perspective, but the relevance of each property to the Meefs system is highlighted. It should be noted that the assessed aspects are relevant to many additional corresponding renovation solutions.

**Table 1.** An assessment of the thermal and moisture properties of the Meefs system.

| Property | Issue to be Considered | Relevance to Meefs | Remarks |
|---|---|---|---|
| **Thermal properties** | | | |
| Performance of the thermal insulation under different temperature and moisture conditions | Stability of dimensions and thermal performance properties | Wide variation of potential materials | |
| Thermal insulation that acts as a tolerance compensating layer filling the gap between the rough surface of the old wall and the technical panel | It should be made of soft material with low compression strength to be able to form uniform contact with the old wall surface | Back layer insulation | The thickness of the layer depends on the requirements of the assembly, i.e. rough surface requires thicker back layer insulation |
| Thickness in cold climates | Insulation material | Thick building envelopes possible depending on the insulation material used [34] | New insulation materials with low thermal conductivity allowing a thin insulation layer are expensive [35–37] |
| Avoiding varying thermal and moisture loads | Insulation of connections | Electrical and hydraulic connections and installations of the operation control system are integrated into the Meefs structure | Varying loads like liquid water contact and freezing should be considered/prevented |
| Cooling effect of the green façade | Not necessarily profitable in cold climates due to good insulation. Seasonal maintenance. | Green façade units can be integrated into the Meefs [33] | Plants need an irrigation system which means additional costs and produces a new source for moisture loads. The humidity influences thermal conductivities [38]. |
| Overheating of the cavity space | Ventilation in order to diminish overheating in summer and contribute to energy savings in the winter [39] | The Meefs system works as the second skin of the building | Some other issues relevant to double skin façades (e.g., [40]) may also be relevant to the Meefs system |
| Adequate cooling of the photovoltaics (PVs) (efficiency decreases under high temperatures) | Mechanical ventilated façades could ensure adequate cooling of the PVs [41] | PV is one potential technology for the Meefs façade | |
| Restricting effects of thermal bridges | Continuous thermal insulation. Effect of structure frames on the U-value. | Panel framing reduces the thermal resistance of the technical units. Back layer insulation is especially important for the U-value of the renovated system. | Thermal conductivity of the framing material and the form of the frames have strong effect on the thermal bridge effect |
| **Moisture properties** | | | |
| External water condensation | Harmful action of moisture [42] in glazing | U-values of glazing below 1.3 W/m$^2$K, where the risk is most potential [43], can occur also with Meefs technical panels | Different window coating solutions reduce this risk [43,44] |
| Adequate drying efficiency of the renovation system to meet with the loads from indoor air and from the (possibly wet) existing structure | Local accumulation of moisture inside the technical panel or on the boundary between the panel and back layer insulation | No ventilation system planned but requires ventilation at least in Northern and Central-European conditions. | Ventilation affects the thermal performance of the wall. The effect depends on the ventilation route and rates. |
| High enough vapor diffusion coefficient of the back layer insulation | Sufficient moisture flow from the old structure | Risk of high local moisture contents if not considered. Also, consider the drying of the new panel system as a whole. | From back layer insulation, the moisture should be ventilated or transported through the panel insulation to the panel ventilation routes. This aspect is only a part of the moisture performance. A holistic approach is needed to evaluate the whole performance. |
| Condensation of water inside the units including phase-change materials (PCMs) (or inside any other panel that does not allow drying of internal moisture loads) | Protection from rain and frost | Two new technical units are based on PCMs | Proper selection of phase-change materials in different climate conditions has a big impact on their functionality [45,46] |

**Table 2.** An assessment of other properties of the Meefs system.

| Property | Issue to be Considered | Relevance to Meefs | Remarks |
|---|---|---|---|
| **Structural properties** | | | |
| Using fiber reinforced polymer (FRP) | Long-term durability in cold climates | The FRP structure hosts all the installations in the Meefs system. | Benefits of FRPs include long-term durability, weathering resistance, exceptional mechanical properties and suitability for prefabrication [47–49] |
| Structural safety | Endurance in weather conditions | Any breakage of the exterior system may cause risks of falling parts to the surroundings. | FRP-based case studies can give more evidence [50] |
| Mechanical strength and structural stability | Clarification of the exact structure | It is not yet clear if the panel framing system is in contact with the existing wall surface | If the frames are supported 140 mm apart from the existing structure the mechanical strength and structural stability aspects have to be taken into account when designing the system. Suitability of the existing structure for the new load bearing requirements. |
| Positioning of the structure | Vertical and horizontal directions and the smoothness of the surface of the old structure is not perfect. How to reach good contact with new system having straight surfaces. | Risk of uncontrolled ventilation routes between the existing wall and the Meefs panel may affect the thermal performance of the system | This aspect is related to the back layer insulation, i.e., smoothing out the irregularities and adjusting the old sloping and the new straight envelope |
| Window openings | Existing windows are left deep from the exterior surface of the new panel/new exterior window structure | The old windows will remain. Their position is not ideal for the thermal or moisture performance due to changes in temperature fields and the possible thermal bridging | Changing windows completely could be considered to overcome this challenge more easily [21] |
| **Fire properties** | | | |
| Tolerance against fire of the thermal insulation | Potentially increased fire load on façades with combustible thermal insulation [51] | Meefs is modular, unlike EPICS, but fire protection needs to be taken into consideration | |
| Fire safety regulations | Requirements differ for ventilated and non-ventilated façades [52] | At least limited ventilation seems to be necessary for the system to have safe moisture performance. Careful selection of thermal insulation materials The fire capacity and other fire properties of the technical units needs to be assessed prior to installation. | Must be defined separately for each building according to regional regulations |
| Tolerance against fire of fiber reinforced polymer (FRP) | Need of experimental fire safety tests [49] Smoke emission under fire needs to be carefully assessed [53] | Used in the structure of Meefs | Pyrolysis simulations are not an effective way to design fire safety of FRPs for architectural applications [49] |
| **Cleaning properties** | | | |
| Cleaning of the supply air in units including PCMs | Air filtering solution in units including PCMs | Valid if replacement air is taken through the unit | Air filters should be easy to change and the air channels easy to clean |
| Cleaning of the ventilation channels | Air flow routes | Some Meefs units are used for fresh air intake or are part of the building ventilation system | The air channels should be easy to clean |
| **Data security properties** | | | |
| Data security | Connection to Internet | Remote control of the operating control system | Unprotected remote management functions in Internet can be used for criminal purposes [54] |

## 2.3. Target Building Stocks

### 2.3.1. Finnish Apartment Buildings

At the end of 2012, 44% of all dwellings in Finland were in blocks of flats [55]. In Figure 2, the Finnish multi-family apartment building stock at the end of 2012 is shown, divided based on the construction year [56]. In total, there are almost 58 000 apartment buildings in Finland with a total gross floor area of about 92.5 million m². The majority of these buildings were built in the 1960s, 1970s and 1980s. In 2012, the average floor area of a dwelling in a block of flats was 56.5 m² [55].

In Finland, nearly 50% of apartment buildings have concrete-based walls. Brick-based walls represents 33% of the constructions [57]. Several concrete wall configurations are described [57–59]: concrete sandwich panels with low thickness insulation, breeze concrete walls, and reinforced concrete and breeze concrete walls. Most typical façade structures in apartment buildings built during 1960–1975 are 3-layer sandwich panels including 50 mm of reinforced concrete, 90 mm of mineral wool insulation and 80/150 mm (depending on whether a non-bearing or a bearing inner layer) of reinforced concrete [59]. Lahdensivu and Hilliaho [60] noticed that the real insulation thickness often varies from the design values, as seen in Table 3. Typical U-values of Finnish non-renovated apartment buildings are presented in Table 4.

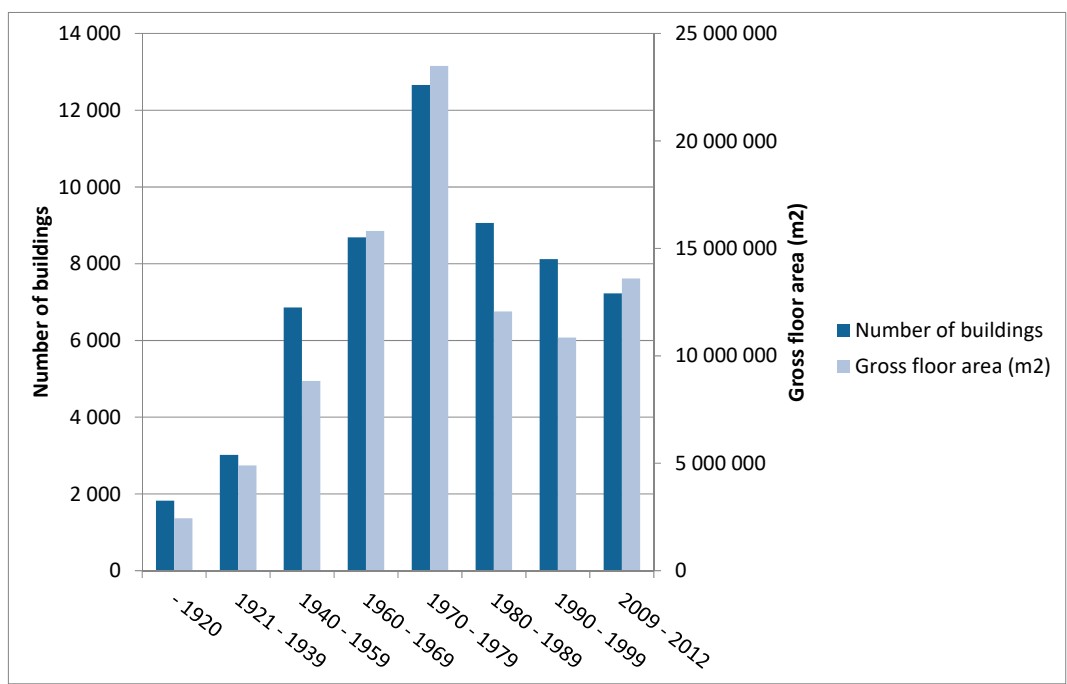

**Figure 2.** The Finnish apartment building stock as of 31.12.2012 [56].

**Table 3.** Design and actual average thicknesses of insulation layers in Finnish apartment buildings based on the year of construction [60].

| Year of Construction | Design Value (mm) | Actual Average Values (mm) |
|---|---|---|
| 1963–1975 | 90 | 83 |
| 1976–1985 | 120 | 109 |
| 1986–1996 | 140 | 131 |

**Table 4.** Typical U-values for non-renovated apartment buildings in Finland and cold climate zones in general.

| U-values in W/m$^2$K | For a Finnish Apartment Building from the 1970s (Nieminen [21]) | For Finnish Apartment Buildings Built in the 1970s (Häkkinen et al. [58]) | For Buildings in Cold Climate Zones Built before 1975 (Lechtenböhmer and Schüring [61]) | Maximum for Buildings in Cold Climate Zones Built after 1975 (Lechtenböhmer and Schüring [61]) |
|---|---|---|---|---|
| façade | 0.3–0.40 | 0.475 | 0.50 | 0.25 |
| roof | 0.3–0.40 | 0.335 | 0.50 | 0.18 |
| floor | N.A. | 0.48 | 0.50 | 0.19 |
| windows | 2.10–2.40 | 2.44 | 3.00 | 1.60 |
| | | N.A. = not available | | |

### 2.3.2. German Apartment Buildings

In Germany, there are over 3 million apartment buildings with a total living space of about 1400 million m$^2$ [29]. These numbers include multi-family houses and apartment blocks, but not terraced houses. Overall, the German apartment building stock is older than the one in Finland. The majority (20%) of German apartment buildings have been built between 1958–1968 (Figure 3).

The dominant structure of the German housing stock is brickwork, which can be found in 64% in the form of one layer and in further 29% in the form of two layers with cavity [29]. Technical or regulatory restrictions for adding extra insulation exist for about half of the old buildings. Table 5 shows average U-values in German housing.

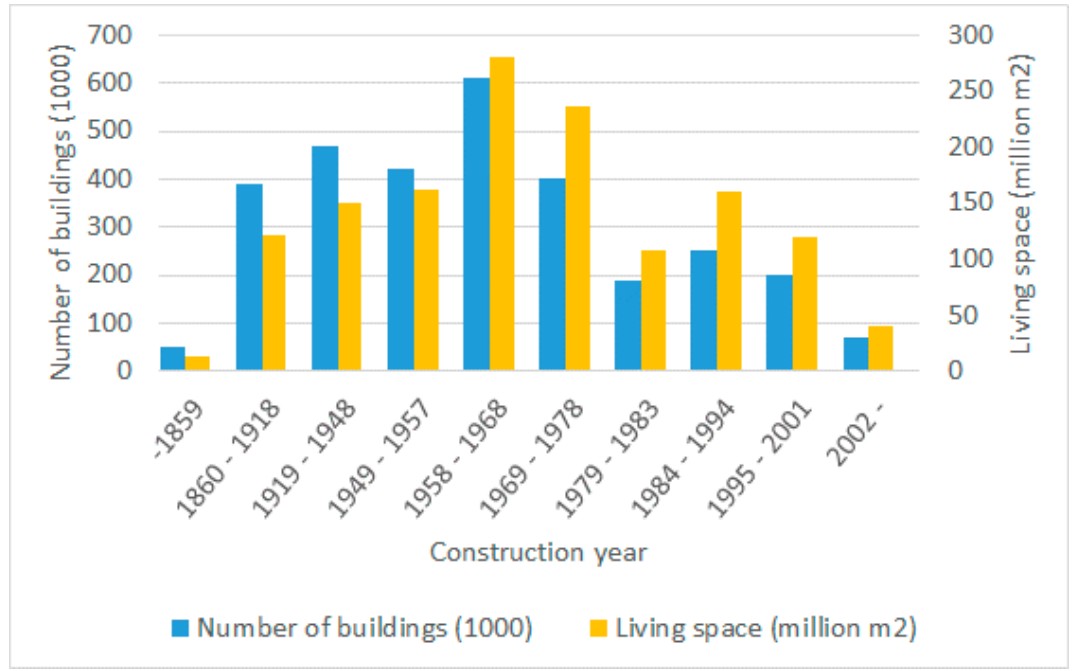

**Figure 3.** The German apartment buildings based on the construction year (data retrieved from [29]).

**Table 5.** Average U-values in German residential buildings (W/m$^2$K) (data retrieved from [29]).

| Construction Year | Wall | Roof | Floor | Window |
|---|---|---|---|---|
| until 1978 | 1.15 | 0.77 | 1.05 | 2.64 |
| from 1979 to 1994 | 0.64 | 0.40 | 0.71 | 2.37 |
| from 1995 | 0.28 | 0.23 | 0.36 | 1.28 |

### 2.3.3. Spanish Apartment Buildings

In Spain, there are about 1.9 million apartment buildings, including over 14.1 million dwellings (Figure 4). Considering that the average floor area of a dwelling in a multi-family building is 62 m$^2$ [62], the total living space in Spanish apartment buildings is about 877 million m$^2$. About 22% of the buildings were built in the 1970s and about 17% in the 1960s.

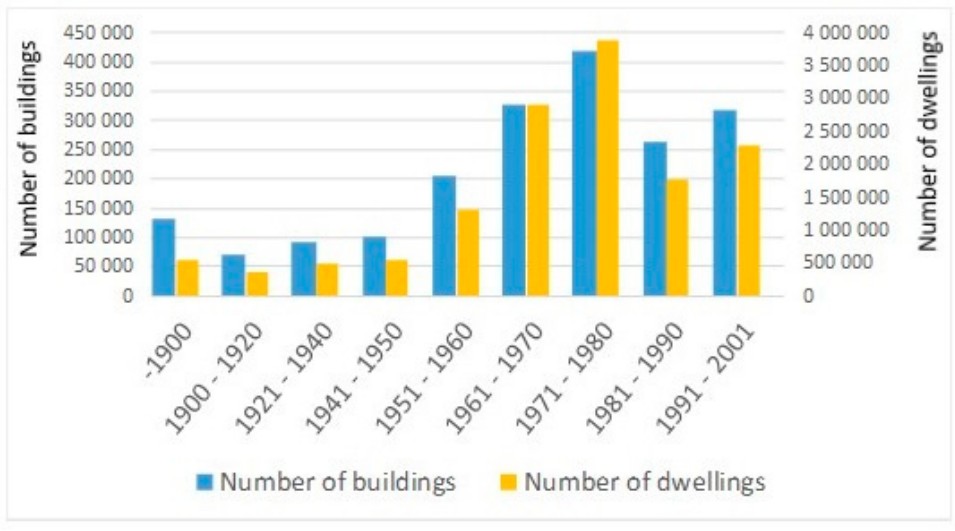

**Figure 4.** The Spanish apartment building stock (data retrieved from [63]).

Spain is divided into several climatic zones where building properties also differ [62]. Table 6 shows typical U-values in the Madrid area.

**Table 6.** Typical U-values in the Madrid area (W/m$^2$K) (data retrieved from [64]).

| Construction Year | Wall | Roof | Floor | Window |
|---|---|---|---|---|
| before 1975 | 1.7 | 2.25 | 2.25 | 5.8 |
| from 1980 to 2005 | 1.2 | 0.9 | 1.2 | 5.8 |
| from 2006 | 0.86 | 0.49 | 0.64 | 3.5 |

## 3. Hygrothermal Performance Challenges of Meefs Retrofitting System

The Meefs system works as a second skin façade, while the original façade is preserved. Therefore, thermal and moisture performance need to be carefully analyzed before installation. In this section, hygrothermal performance issues of the Meefs system are introduced based on own expertise. For example [15,65,66] discuss principles of façade-related building physics in general.

Structures should be designed so that they do not accumulate moisture and the possible seasonal moisture increase remains on a safe level for the materials and systems. Structures should be able to dry out from typical moisture loads from indoor and outdoor air and also from the initial building moisture that the materials may contain after the construction or renovation. In cold and mild climates, the drying may take place mainly towards outdoor air or both to indoor and outdoor air, and in hot climates mainly towards the indoor air.

The moisture transport out from the structure can be based on water vapor diffusion through the material layer and surfaces and also on the possible ventilation of the structure.

Special challenges for the hygrothermal performance can be encountered in Meefs systems that have vapor impermeable exterior surfaces, like photovoltaic (PV) panels, and in some phase change material and active façade applications. In climates where the drying of structures to outdoor air is dominant, the closed technical elements form severe risks for moisture accumulation. In these cases, the drying efficiency of the system should be considered, designed and applied properly. Ventilation of the system with outdoor air could be one solution when the diffusive moisture transport out from the structure is reduced. The ventilation could be limited through local channels in the structure or it could be based on a continuous ventilation cavity when higher drying efficiency is needed.

Structure ventilation always has some effect on the thermal performance of the system, depending on the ventilation route and rate. High ventilation inside the structure strongly reduces the effect of the thermal resistance of the layers that are outside the ventilation cavity. The U-values solved for a non-ventilated case are not valid when ventilation is applied.

### 3.1. Thermal Performance—Effect of Non-Ideal Contact Conditions and Air Cavities

In the energy saving calculations, it was assumed that the renovation panel system forms a uniform, new part of the wall system. The principle assumption was that the thermal resistance of the new exterior structure layers can be fully included in the improvement of the U-value of the existing wall. However, it is difficult to reach a perfectly uniform contact with the renovation panels and the old wall surface. The old walls are not always perfectly straight, the surfaces can be uneven, and some structural details may protrude from the average wall level. These are challenges that most likely lead to non-ideal contact conditions, i.e., air cavities in the renovated structure system.

The thermal insulation between the technical panel and the old wall is typically mineral wool that allows relatively smooth contact with the existing wall. Mineral wool is very air permeable, so any air cavity in the boundary between the Meefs system and the existing wall can increase air convection inside the structure and decrease the effect of the new thermal insulation layer. Special attention has to be paid to the design and assembly of the back layer insulation to minimize the possible convection in this boundary.

If there is any air flow between this cavity and the outdoor air, this air flow causes heat losses that may considerably reduce the effectiveness of the additional thermal insulation. Figure 5 presents the critical contact conditions and possible convection routes in the structure renovated with the Meefs panel system.

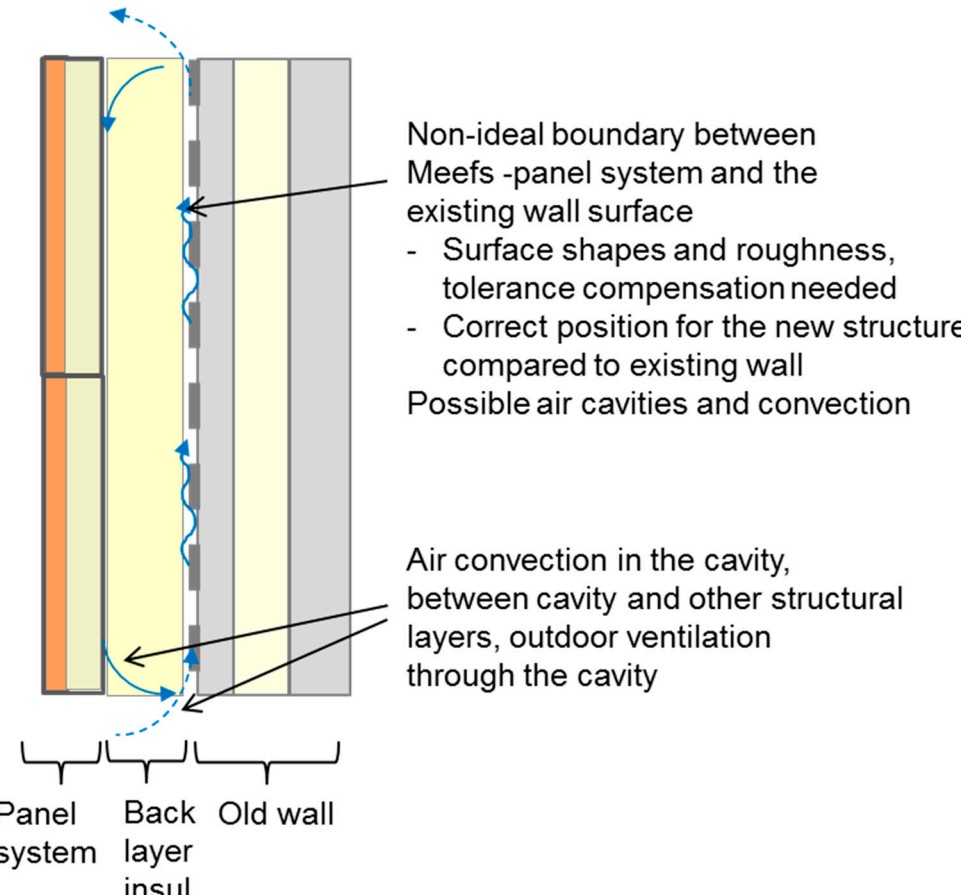

**Figure 5.** Non-ideal contact between the structure elements of the Meefs panel system and the existing wall may result in uncontrolled ventilation in the structure. This may affect the thermal and moisture performance of the system.

If the Meefs panel system includes a ventilation solution, the real U-value of the renovated wall is strongly dependent on the place of the ventilation cavity and the ventilation volume. If there is strong free outdoor air convection in a continuous ventilation cavity, the exterior panel systems act merely as protection against wind and thermal radiation. In this case, the thermal insulation that is on the exterior side of the ventilated cavity has practically no effect on the overall U-value of the wall system. The additional thermal resistance of the panel system would be in this case in the range of 0.2 (K m$^2$)/W. When the wall ventilation is more limited, the exterior thermal insulation layers have more effect on the U-value, but still less than in a non-ventilated case. The challenge is to have adequate ventilation to remove the additional moisture and to ensure that the new panel improves the thermal performance of the wall.

From the thermal performance point of view, any structure ventilation should be avoided, and the thermal insulation layers should be protected against wind and convection by using airtight wind barrier layers. However, to have safe moisture performance, the Meefs system should be able to dry out the additional moisture from the system. However, this has not been taken into account in the energy performance analysis of the renovation system. When the moisture has to be dried out using structure ventilation, the thermal performance changes from the simplified ideal case assumptions as

well. Figure 6 presents the critical parts of the Meefs system when evaluating the moisture performance of the renovated façade.

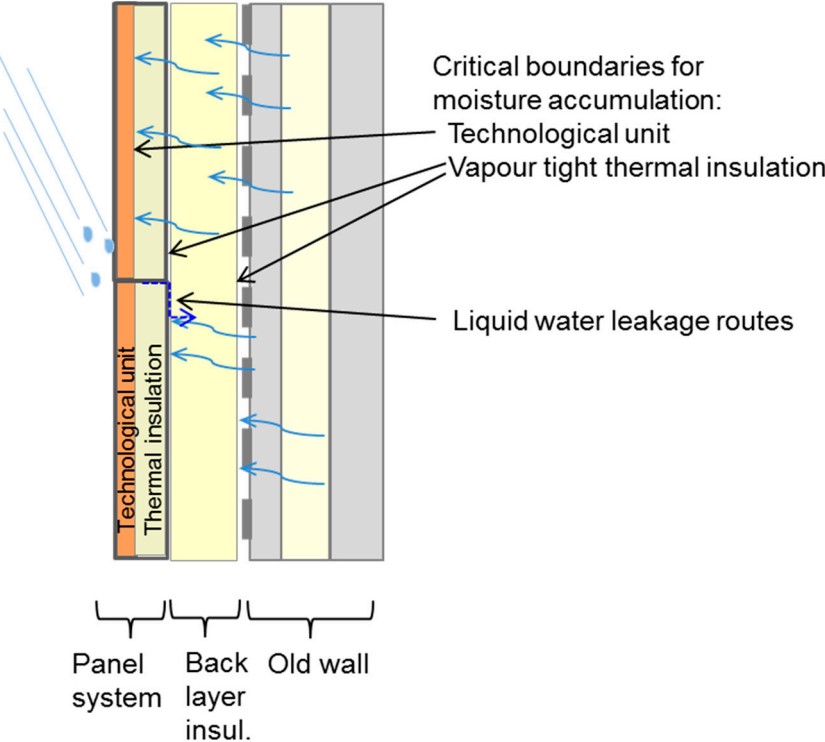

**Figure 6.** Critical parts for the moisture performance of the Meefs panel system.

When the new façade elements form a practically vapor tight layer on the exterior side of the structure, wall ventilation should take care of the drying of moisture from the structure. Otherwise there is a risk of moisture condensation and accumulation on the inside surface of the exterior layers. This can cause deterioration of materials and systems, and it affects the thermal performance. In the worst case, the wet materials affected by changing temperature conditions can permanently lose their performance properties. The ventilation should correspond to the moisture loads, but in order to have any use of the new thermal insulation at the exterior side of the ventilation cavity, the ventilation flow should be as low as possible.

There are three main moisture load sources in the structure that should be taken into account in the design of the system:

- Load 1) Initial moisture of the material layers, accumulated mainly in the exterior parts of the existing wall (exterior concrete core, brickwork, etc.). Typically, the driving rain is the main reason for the high moisture content.
- Load 2) Moisture flow from indoor air into the structure. This takes place typically by diffusion, and in some cases the air leakages between indoor air space and structural layers can also enhance local moisture transport into the wall.
- Load 3) Possible water leakages through the exterior parts of the wall, typically defects in the structural details or weather protection layers.

Load 2 from indoor air is typically used as the design load for structures. However, in the renovation cases, Load 1 may highly exceed this load, and the renovation system has to be designed so that it can dry out the initially wet old structure. Load 3 cannot be precisely quantified, but these risks have to be taken into account in the design of the new system.

The new thermal insulation layer should have high enough vapor diffusion coefficient to enable sufficient moisture flow from the old structure towards the outer layers and out of the structure.

A vapor barrier that is too tight can cause moisture accumulation in this first boundary between the Meefs system and the existing wall structure.

As a summary, the U-value of a façade renovated using a partly ventilated renovation system cannot be solved using simple addition of the thermal resistances of the old wall structure and the new panel components. When evaluating the thermal performance or energy saving potential of this system, the ventilation flow rates of the structure have to be also considered.

### 3.2. Air Tightness of the Renovated Façade

One aim is to improve the air tightness of the building envelope during renovation. When the overall air tightness of the building envelope can be improved, energy efficiency also improves. The uncontrolled air leakages through the structures decrease, and it is easier to control the pressure fields and ventilation rates of the indoor spaces. Additionally, thermal comfort may increase due to reduced draught.

The improved air tightness can cause risks for the ventilation of indoor spaces. If the surplus air for the room has been taken mainly through the leakage routes of the exterior wall and window frames, there is a risk of too low ventilation in the room space when the ventilation openings to outdoor air are reduced in the renovation. This risk is high when natural or exhaust ventilation is applied. Especially with exhaust ventilation the indoor under-pressure may increase due to the improved air tightness of the façade. This may increase the air leakages through the parts of the building envelope that contain contaminants, which affects the indoor air quality.

Proper ventilation of the room space has to be ensured when applying new structural elements that may change the air leakage properties of the building envelope.

### 3.3. Thermal Bridges in the Renovation System

When there is an air cavity (ventilated or closed) in the new façade element, the air flow tends to equalize the temperature levels in the cavity. There is, of course, some temperature distribution due to the forced or natural ventilation flow, but the temperature levels are typically relatively uniform. If the thermal resistances of the new façade elements differ significantly from each other, these differences cause local thermal bridges affecting the thermal performance of the whole system. In addition, the possible thermal bridges of the framing system could be reduced by using uniform thermal insulation layer. Any non-uniform thermal resistance of the system dilutes the overall thermal performance of the assembly.

Windows and the exterior window elements have a different thermal performance than the opaque wall elements due to typically lower thermal resistances and additional heat gains from solar radiation. The framing system itself causes thermal bridges that may affect the thermal performance of the adjacent insulated elements. The overall thermal performance of the window element and the possible effects on the adjacent elements have to be taken into account in the design phase.

The sealing and protection against driving rain loads is one critical aspect for the whole moisture performance of the system. The window openings are typical risk areas for water penetration into the structure. The same risks are valid also for the panel system. The design and installation of the exterior glazing and frame systems, especially if the windows are meant to be opened, have to take into account the pressure differences and driving rain loads caused by the possible maximum wind velocities in the area. The height of the building also has a significant effect on these loads, as the driving forces increase in tall buildings, and the water running down the exterior surface increases the local loads.

### 3.4. Moisture Performance Principles

When developing new façade solutions, one key issue is to ensure safe moisture performance of the structure after the installation of the new elements. This includes several points that have to be considered in the design and installation processes. The redistribution of the initial moisture of the old structure can especially cause severe risks for the renovated façade system. Different climate

conditions bring additional challenges in the moisture performance requirements: Cold climates may set different requirements for the structures than mild or warm outdoor conditions.

In the presented Meefs renovation system, the new technological module is installed on the exterior side of an existing wall. This multifunctional panel may have a totally vapor tight exterior layer, like PV-panel, glazing, etc. In addition, no moisture performance plan, like a structure ventilation scheme, has been presented for the system. This means that the initial additional moisture from the existing wall and that of the new system could be dried out only inwards. However, this is not possible under cold outdoor conditions when the temperature and partial vapor pressure gradients drive the moisture towards exterior parts. In these cases, the possible additional moisture accumulates into the colder exterior parts of the structure, where it can damage the materials or technical systems of the new modules. The damage mechanisms that overly high local moisture contents can cause are, for example, freezing and freeze/thaw cycles, corrosion, exceeding critical moisture contents of the materials, mold growth, malfunction of electrical systems (PV-cell systems), etc. In a sustainable system, the structures should be able to dry out without having problems caused by locally accumulating moisture.

## 4. Numerical Analysis of the Hygrothermal Performance of Retrofitted Buildings

In this section, case analyses are made in three different climatic locations, namely in Vantaa in Southern Finland, in Holzkirchen in Southern Germany and in Madrid in Spain. These locations were selected to cover some typical climate conditions in the cold northern, mild central and warm southern parts of Europe, acknowledging that different climate zones may exist in a single country.

### 4.1. Solutions for Safe Moisture Performance of Exterior Renovation Systems

There are some options how the exterior renovation panel system with additional thermal insulation system can have a safe moisture performance:

### 4.1.1. Vapor Open Exterior Layers

The panel system with thermal insulation layers and the exterior layer (the technological unit) should be open for diffusion so that the additional moisture can dry out.

If some parts of the panel system are vapor tight, like PV-panels and other such technological units, the adjacent vapor open parts can partly compensate for this. The requirement is that moisture can enter sideways freely into these vapor open parts and that the area of the vapor tight parts is limited enough so that diffusion to adjacent areas is possible. In the presented case, this solution did not seem to be possible due to the high area of vapor tight exterior surface elements and these solutions were not analyzed numerically.

### 4.1.2. Ventilation of the Wall

The technological units are typically vapor tight, and these units can form a significant part of the wall area. In this case, the only possibility to ensure moisture safe performance is to have ventilation in the structure. Ventilation of the Meefs panel system with outdoor air should cover the whole renovated façade area, and the ventilation flow route should be as close to the exterior surface as possible. When designing wall ventilation of the framing system, the placement of the thermal insulation layers and the ventilation flow routes and rates have to be taken into account to ensure good thermal and moisture performance. Ventilation may also set special requirements for the fire safety of the materials.

Two possible ventilation schemes of the renovation system were studied:

(1) Ventilation scheme 1

Limited ventilation of the renovation panel. The wall ventilation may cover only part of the wall area. The ventilation routes may be local ventilation channels (edge area of the frames) or grooves in thermal insulation that form a ventilation network close to the exterior surface of the technical module.

(2) Ventilation scheme 2

Continuous air cavity between material layers. In this case, the possible boundary to have a ventilation cavity would be between the technical module and the back layer thermal insulation that is used to ensure firm assembly of the panel system on the existing wall surface. In this case, thermal resistance of the exterior insulated panel will be reduced.

Both of the ventilation cases (presented in Figure 7) were studied using a well-known, commercial, numerical simulation software WUFI 6.1 [67]. WUFI® allows realistic calculation of the transient coupled heat and moisture transport in multi-layer building components exposed to natural weather [67]. WUFI® is also widely used in the scientific community. For example, [68–70] have utilized it in various building physics analyses.

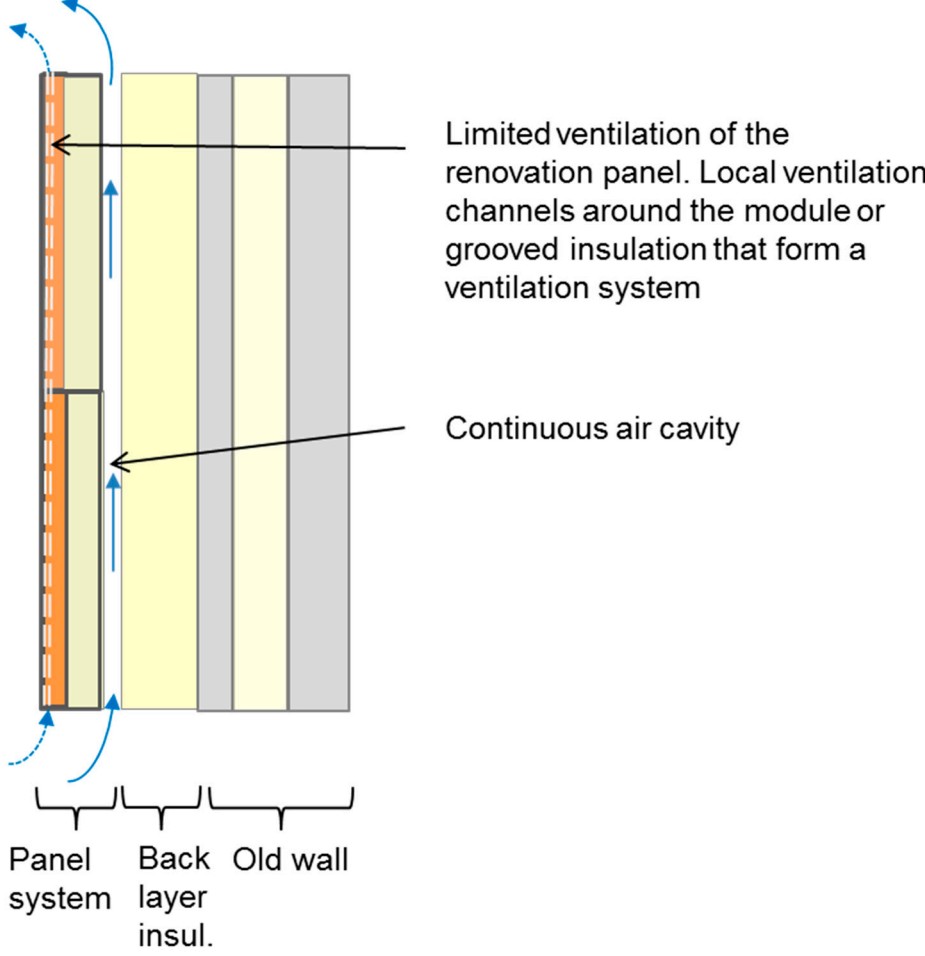

**Figure 7.** Two possible ventilation schemes of the Meefs renovation module system.

In the scheme with ventilation only on the edges of the panel, the ventilation was modeled as one uniform (10 mm) thin air layer having reduced ventilation. This analysis shows the effect of ventilation per wall surface area (dm$^3$/s/m$^2$) regardless of the ventilation scheme—whether it is realized using continuous ventilation cavity, grooved thermal insulation or a network of local ventilation channels. The difference between the cases is in the ventilation air flow rates—in the limited ventilation scheme the air flow rate and capacity to remove moisture by ventilation is quite limited.

*4.2. Analyzed Cases of the Wall Ventilation*

Two old wall structures were studied:

B)　Brick wall with 130 mm of brick work on both sides of 100 mm thick mineral wool insulation.

C)    <u>Concrete</u> sandwich structure with 80 mm exterior concrete core, 100 mm of mineral wool insulation and inside 120 mm concrete layer.

Both structures were assumed to be renovated using the presented renovation module consisting of the module having 50 mm of polyurethane (PU) insulation (with vapor permeable surfaces) and 80 mm of mineral wool as the back layer insulation between the old wall and the new panel system. The simulations start from the installation of the renovation module.

Two assumptions were used for the initial moisture content:

N)    <u>(Normal dry)</u> All the material layers had moisture content corresponding to 80% relative humidity (RH) conditions. This corresponds to the upper limit of safe moisture content. In this case the moisture content of the existing wall structure has to be measured and initial drying has to be possibly carried out before the installation of the renovation system.

W)    <u>(Wet)</u> The exterior brick work or concrete core layer of the old structure had increased moisture content due to driving rain. The moisture content of this layer was set to correspond to 95% RH conditions. All the other material layers had initial moisture content corresponding to 80% RH. This corresponds to the case where the existing structure is not dried out before the installation of the renovation system.

The moisture performance of the renovated walls was analyzed numerically. The simulations were carried out using 1-dimensional intersection of the structures including the new renovation module and the possible ventilation either on the outer part of the renovation module or between the panel and the back layer insulation (Figure 7). The simulations were done using three different climate conditions: Vantaa (Finland), Holzkirchen (Germany) and Madrid (Spain) [68,69]. The outdoor conditions were updated every hour using the climate data for the region, based on the WUFI library [67]. The indoor temperature could vary in the range of +20 +25 °C and relative humidity in the range of 30–60% RH (Figure 8), based on typical values considering thermal comfort in winter.

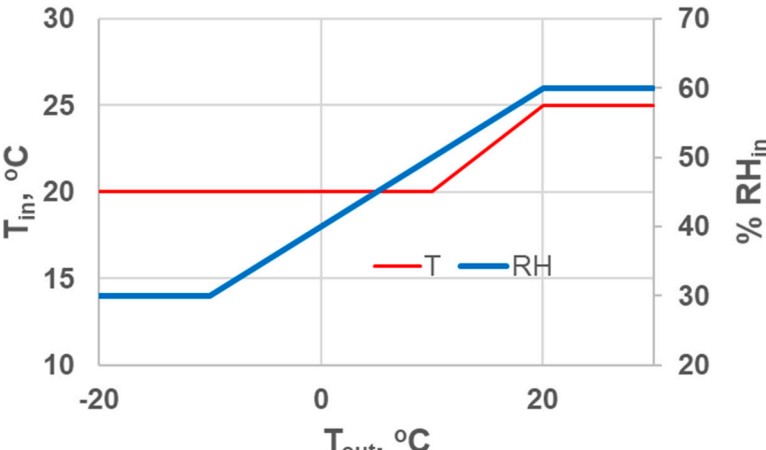

**Figure 8.** Indoor conditions used in simulations. $T_{in}$ = indoor temperature $T_{out}$ = outdoor temperature RH = relative humidity.

The relative humidity values of the ventilated air and that in the exterior side of the back layer mineral wool insulation were used to represent the moisture performance of the wall in different cases. Long period water vapor saturation conditions (100% RH) refer to local moisture accumulation, which is a sign for possible risks in the assembly. Biological growth may start under lower humidity conditions, but this risk was not considered.

The air change rate of the ventilation space (assumption of 10 mm thick continuous cavity) is presented as figure n [1/h] that show how many times in hour the air in the ventilation air space is changed with outdoor air. For example, n = 10 means that the air in the 10 mm space is changed 10 times in an hour and thus the ventilation air flow is 0.03 $dm^3/sm^2$.

Table 7 presents the numerically analyzed cases for ventilation scheme 1 (limited ventilation of the renovation panel) and Table 8 cases for ventilation scheme 2 (ventilation between panel and back layer insulation). The results for ventilation scheme 1 studies are presented in chapter 4.3 and those for scheme 2 in chapter 4.4. The cases presented for ventilation scheme 2 were selected to represent those that have the highest requirements for drying efficiency. The case codes include letter V (Vantaa, Finland), H (Holzkirchen, Germany) or M (Madrid, Spain). Additionally, the codes include n + number referring to the air change rate of the wall ventilation system. In cases where the initial moisture content of the exterior part of the old structure is wet (95% RH), the code has also a letter W. This represents the worst case where moisture from the old structure can be dried.

**Table 7.** Numerically analyzed cases for ventilation scheme 1.

| Case Code | Climate | Initial Conditions Wet/Dry | Air Change Rate in 10 mm Continuous Air Cavity n, 1/h | Old Wall: Concrete (C) or Brick (B) |
|---|---|---|---|---|
| V_n0 | Vantaa, Finland | Dry | 0 | C and B |
| H_n0 | Holzkirchen, Germany | Dry | 0 | C and B |
| M_n0 | Madrid, Spain | Dry | 0 | C and B |
| V_n10 | Vantaa, Finland | Dry | 10 | B |
| H_n10 | Holzkirchen, Germany | Dry | 10 | B |
| M_n10 | Madrid, Spain | Dry | 10 | B |
| V_n10_W | Vantaa, Finland | Wet | 10 | C and B |
| H_n10_W | Holzkirchen, Germany | Wet | 10 | C and B |
| M_n10_W | Madrid, Spain | Wet | 10 | C and B |
| H_n50_W | Holzkirchen, Germany | Wet | 50 | C and B |
| M_n50_W | Madrid, Spain | Wet | 50 | B |
| V_n50_W | Vantaa, Finland | Wet | 50 | C and B |

**Table 8.** Numerically analyzed cases for ventilation scheme 2.

| Case Code | Climate | Initial Conditions Wet/Dry | Air Change Rate in 10 mm Continuous Air Cavity n, 1/h | Old Wall: Concrete (C) or Brick (B) |
|---|---|---|---|---|
| V_n10_W | Vantaa, Finland | Wet | 10 | B |
| H_n10_W | Holzkirchen, Germany | Wet | 10 | B |
| M_n10_W | Madrid, Spain | Wet | 10 | B |
| H_n50_W | Holzkirchen, Germany | Wet | 50 | B |
| M_n50_W | Madrid, Spain | Wet | 50 | B |
| V_n50_W | Vantaa, Finland | Wet | 50 | B |

*4.3. Limited Ventilation of the Renovation Panel*

In the ventilation scheme 1 (Figure 7) the limited ventilation of the renovation panel takes place through the local ventilation channels or grooves in thermal insulation that form a ventilation network on the technical module. Thus, the ventilation flow rate is typically quite limited. In this case, the old wall was an insulated brick wall that represents higher indoor moisture load into the structure than the more vapor tight concrete sandwich wall. In addition, the moisture transport from the brick layer is faster than from the concrete, and it thus requires higher drying efficiency of the renovated system.

4.3.1. Normal Dry Initial Moisture Content

Figure 9 presents the relative humidity values in the ventilation air space (presented as 10 mm continuous cavity) and Figure 10 in the boundary between mineral wool back layer insulation and the PU-insulation of the panel in a case with no ventilation (n = 0) and the initial moisture content corresponding to normal dry (N) conditions. Figures 11 and 12 present the results for a case with ventilation rate n = 10.

Under cold and mild climates (Finland and Germany), even the low initial moisture tended to be transported towards the exterior layers, and the relative humidity on the exterior side of the panel

insulation (air cavity with no ventilation) has moisture accumulation and long saturation conditions (Figures 9 and 10). Only under warm climate conditions (Madrid) the moisture was able to dry out towards the indoor air.

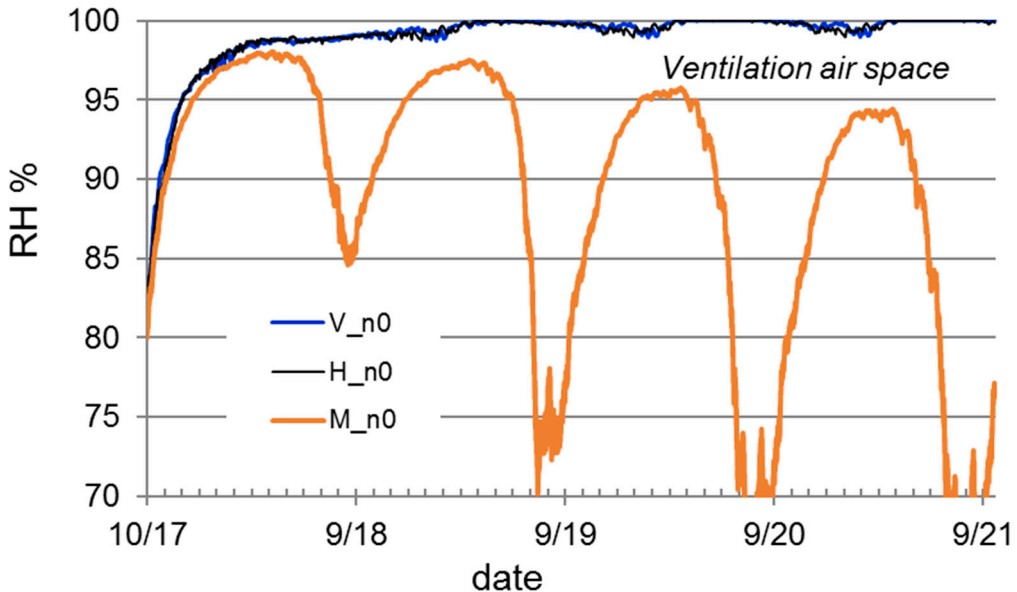

**Figure 9.** Brick wall, no ventilation, normal dry initial conditions (80% relative humidity (RH)). The relative humidity in the ventilation air space in three different climates during the four-year simulation period.

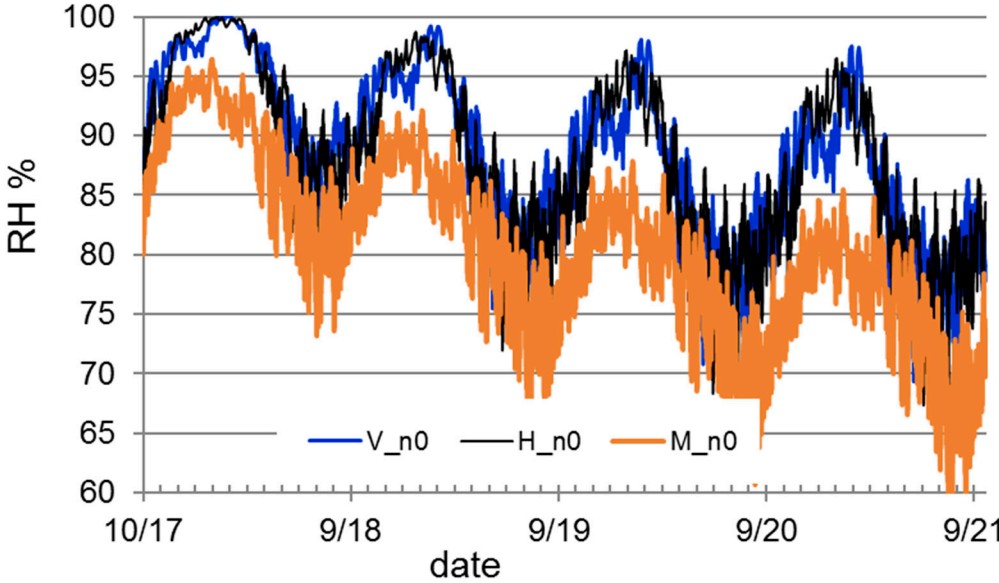

**Figure 10.** Brick wall, no ventilation, normal dry initial conditions (80% RH). The relative humidity in the boundary between mineral wool back layer insulation and the polyurethane (PU)-insulation of the panel in three different climates during the four-year simulation period.

The results show that the renovation module cannot be applied without proper drying ability of the system. This typically means ventilation of the renovation module.

When the air change rate was 10 1/h (Figures 11 and 12), there were only short period condensation conditions under Northern and Central European climate conditions on the boundary of the thermal insulation layers. When the module system has even limited ventilation (10 1/h), it can be assembled on the surface of dry wall structures without causing risk of moisture accumulation on the condition

that the moisture content of the wall is low enough, corresponding to maximum 80% RH conditions. In renovation, the initial moisture level should be ensured by measurements along the drying of the structures under sheltered conditions. In addition, the moisture load from indoor air should not exceed that used in the analysis.

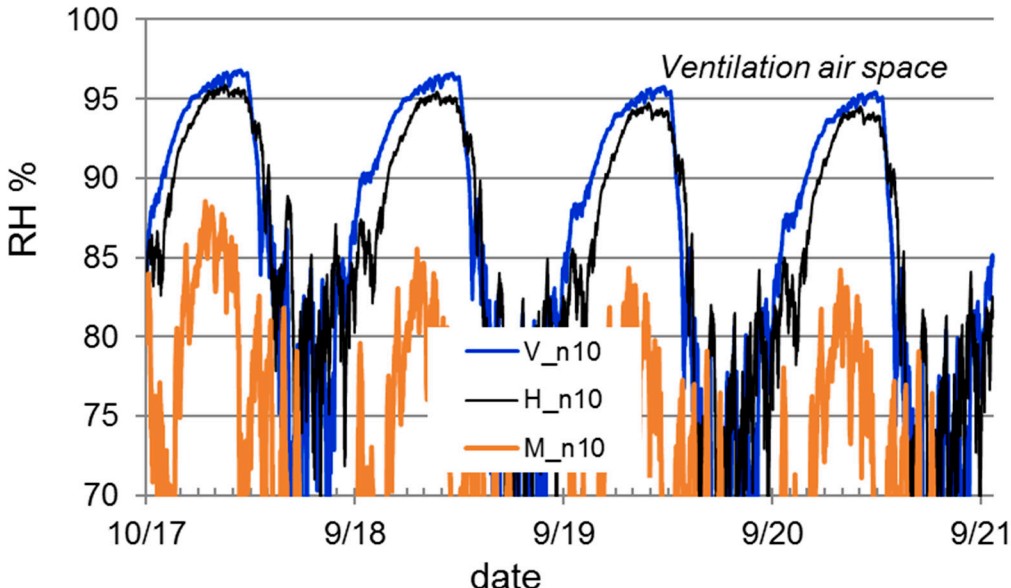

**Figure 11.** Brick wall, ventilation scheme 1, n = 10 1/h, normal dry initial conditions (80% RH). The solved relative humidity in the ventilation air space in three different climates during the four-year simulation period.

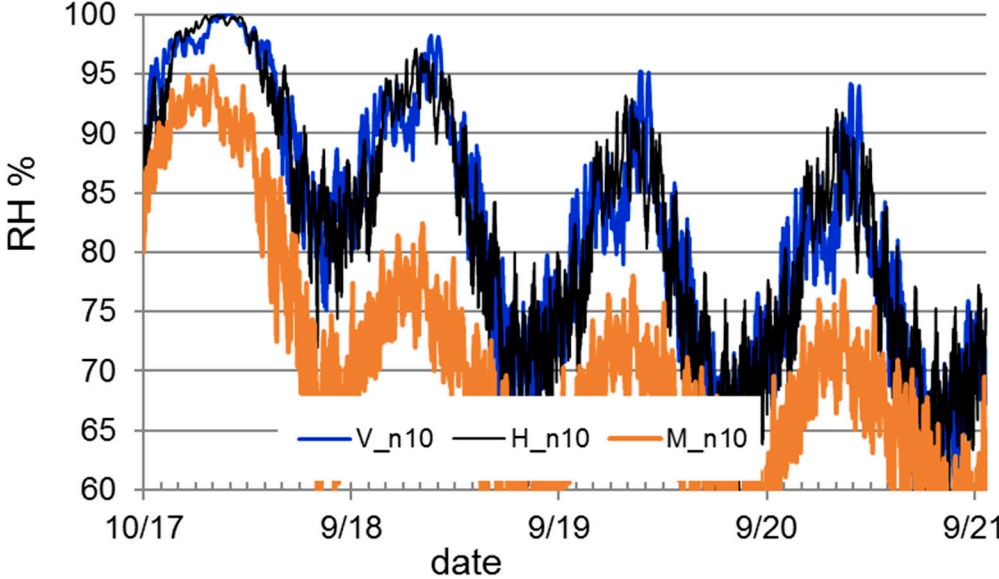

**Figure 12.** Brick wall, ventilation scheme 1, n = 10 1/h, normal dry initial conditions (80% RH). The solved relative humidity in the boundary between mineral wool back layer insulation and the PU-insulation of the panel in three different climates during the four-year simulation period.

4.3.2. Initially Wet Brick Walls

In these cases, the exterior brick layer was assumed to be wet, and the initial moisture content of that layer was 63 kg/m$^3$ corresponding to 95% RH conditions. The drying of the initial moisture from the wall sets requirements for the drying efficiency of the renovation panel system.

Figures 13 and 14 present the relative humidity values in the ventilation air space (presented as 10 mm continuous cavity) and in the boundary between mineral wool back layer insulation and the PU-insulation of the panel in a case with ventilation air change rate n = 10 1/h.

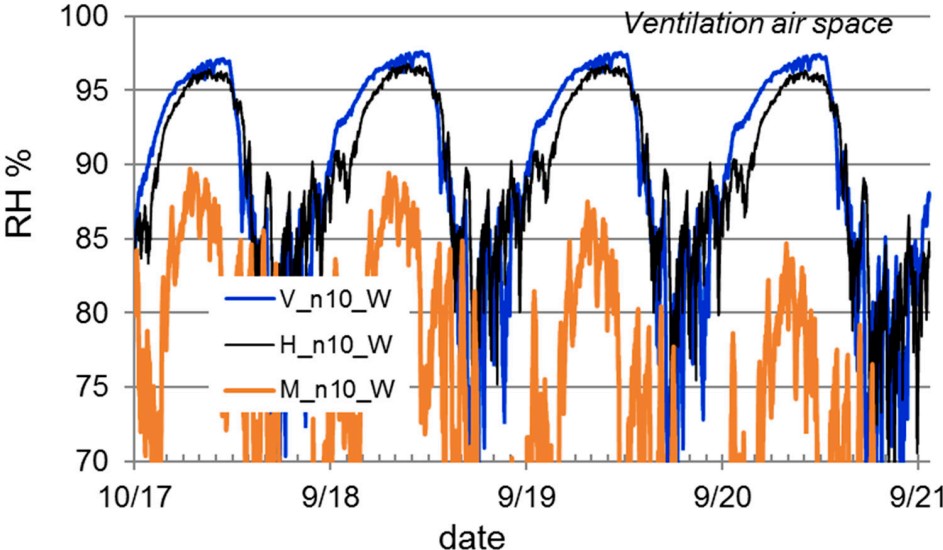

**Figure 13.** Brick wall, ventilation scheme 1, n = 10 1/h, initially wet exterior core of the old brick wall. The relative humidity in the ventilation air space in three different climates during the four-year simulation period.

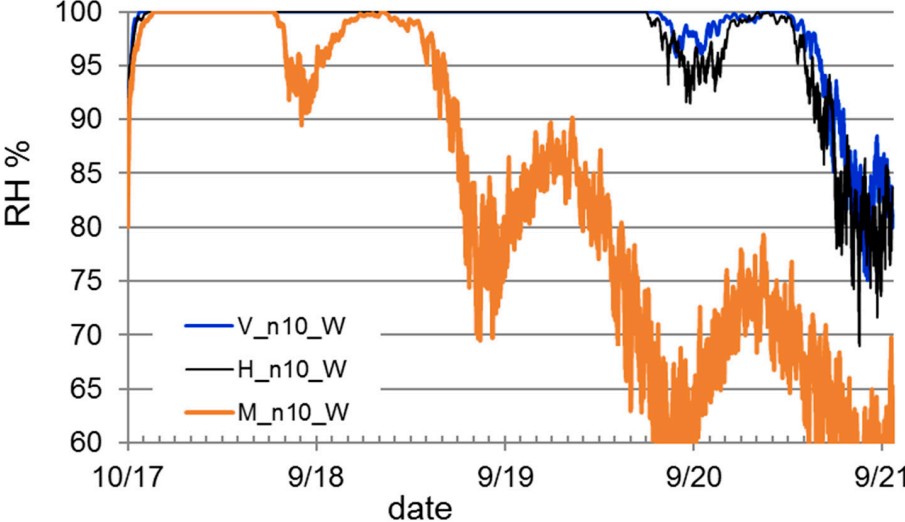

**Figure 14.** Brick wall, ventilation scheme 1, n = 10 1/h, initially wet exterior core of the old brick wall. The relative humidity in the boundary between mineral wool back layer insulation and the PU-insulation of the panel in three different climates during the four-year simulation period.

Figures 15 and 16 present the results in the case n = 50 1/h. Wall ventilation air change rate n = 50 1/h is a very high value for the simulated ventilation system that consists of separate ventilation channels. It represents a high theoretical value instead of a practical case for such a ventilation scheme, and it shows the effect of the ventilation rate on the moisture performance of the renovated wall system.

In both wall ventilation rate cases (n = 10 1/h and n = 50 1/h) there were long periods of condensation conditions in the boundary between the vapor open back layer insulation (mineral wool) and the PU-insulation of the renovation panel. In cold and milder climates (Vantaa, Finland and Holzkirchen, Germany), the condensation was over 2.5 years long, and even in a warm climate

(Madrid, Spain) it continued for about 9 months. The ventilation did not have much effect on the conditions in the boundary, only on the conditions in contact with the ventilation air space.

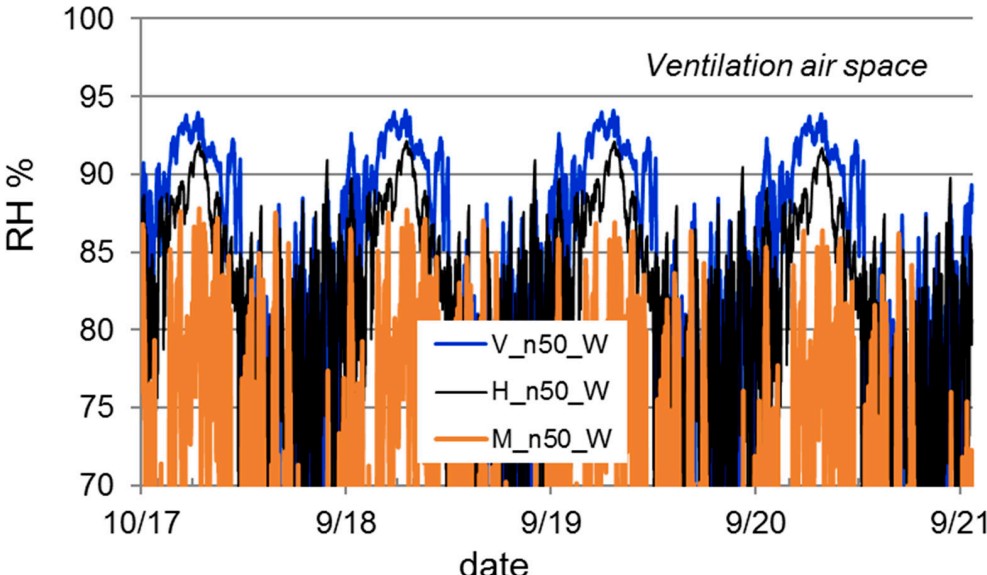

**Figure 15.** Brick wall, ventilation scheme 1, n = 50 1/h, initially wet exterior core of the old brick wall. The relative humidity in the ventilation air space in three different climates during the four-year simulation period.

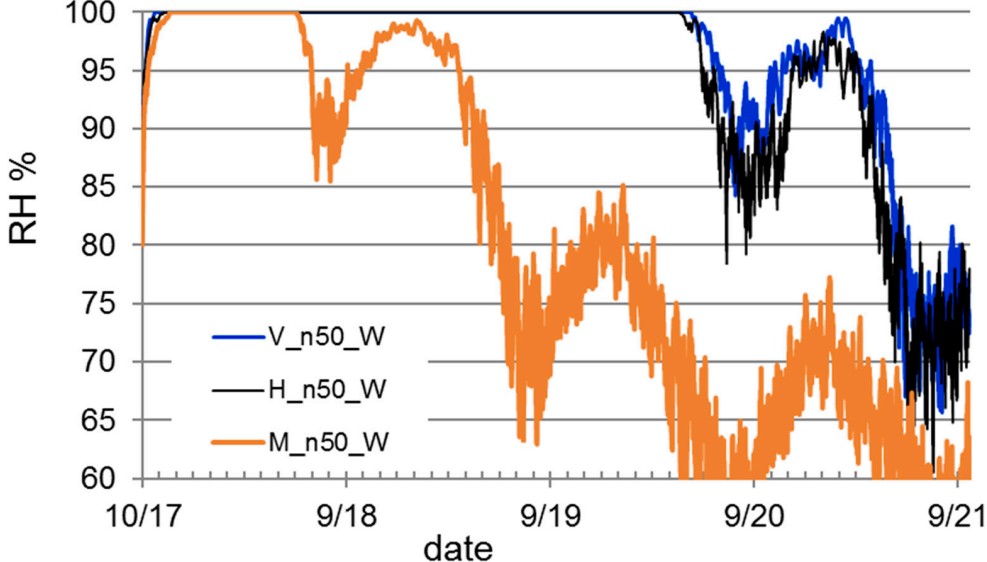

**Figure 16.** Brick wall, ventilation scheme 1, n = 50 1/h, initially wet exterior core of the old brick wall. The relative humidity in the boundary between mineral wool back layer insulation and the PU-insulation of the panel in three different climates during the four-year simulation period.

The long condensation conditions inside the renovated structure system show that the renovation system does not have safe moisture performance, not even with the higher studied ventilation rate. When the ventilation takes place only in the exterior parts of the renovation panel that is insulated from inside with relatively vapor tight PU-layer, the renovated structure system is not able to dry out from additional moisture in the structure without long interstitial condensation conditions. The risks are present in all European climate conditions.

This ventilation scheme can be safely applied only if the brick wall structure is dried out before the assembly of the renovation module to moisture contents below 80% RH level.

### 4.3.3. Initially Wet Concrete Sandwich Walls

In these cases, the exterior concrete layer was assumed to be wet and the initial moisture content of that layer was 118 kg/m$^3$ corresponding to 95% RH conditions. The drying of the initial moisture from the wall sets requirements for the drying efficiency of the renovation panel system.

Figures 17 and 18 present the relative humidity values in the ventilation air space (presented as 10 mm continuous cavity) and in the boundary between mineral wool back layer insulation and the PU-insulation of the panel in a case with ventilation rate n = 10 1/h. Figures 19 and 20 present the results in the case n = 50 1/h for Northern and Central European climates. In addition, in this case the wall ventilation air change rate n = 50 1/h is a more theoretical (too high for practical solutions) value for the simulated ventilation system that consists of separate ventilation channels.

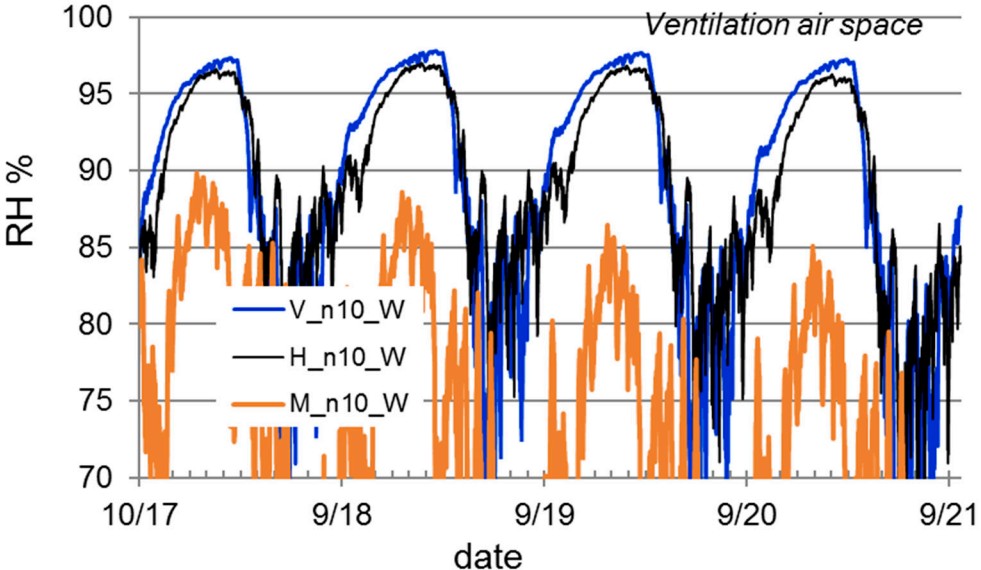

**Figure 17.** Concrete sandwich wall, wet exterior core, ventilation scheme 1, n = 10 1/h. The solved relative humidity in the ventilation air space in three different climates during the four-year simulation period.

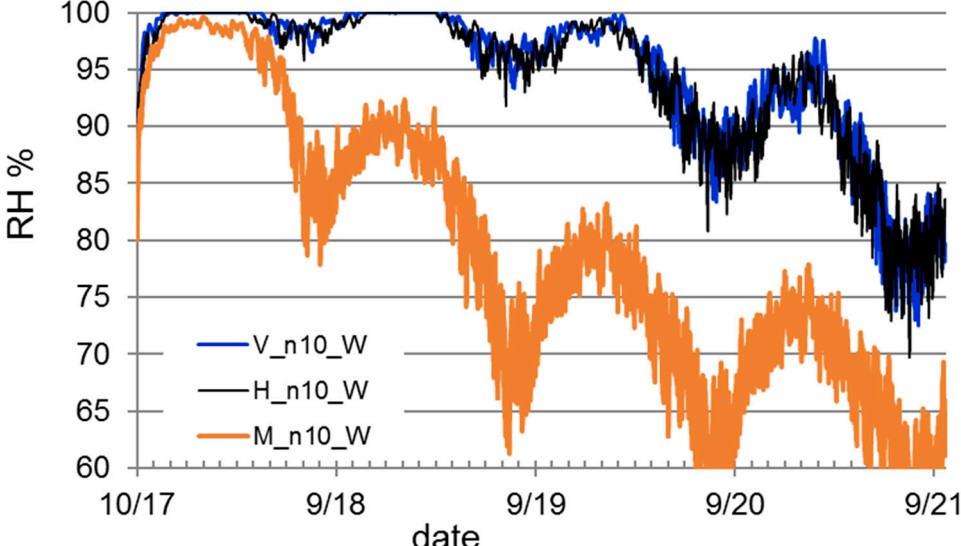

**Figure 18.** Concrete sandwich wall, wet exterior core, ventilation scheme 1, n = 10 1/h. The solved relative humidity in the boundary between mineral wool back layer insulation and the PU-insulation of the panel in three different climates during the four-year simulation period.

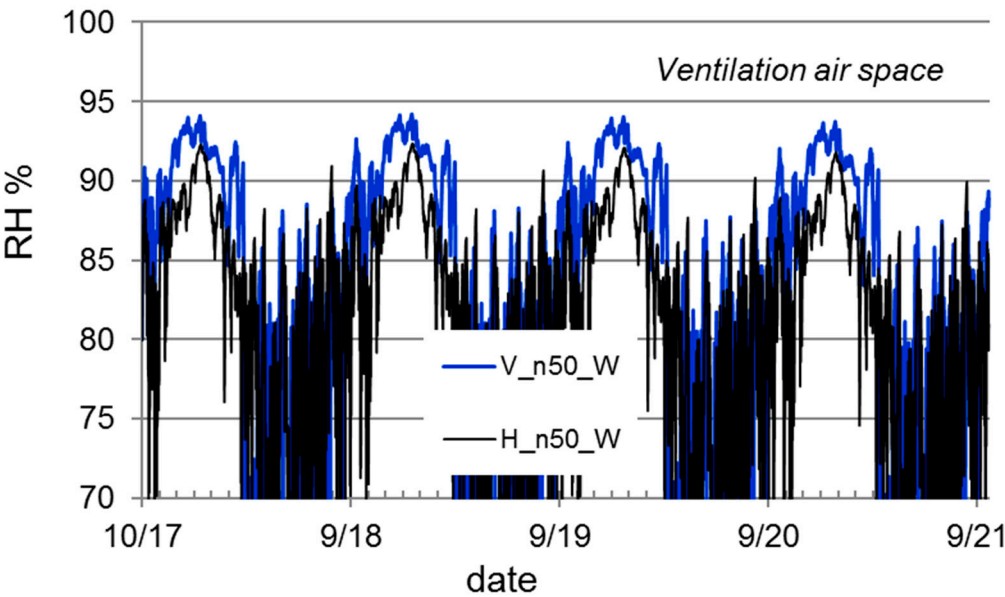

**Figure 19.** Concrete sandwich wall, wet exterior core, ventilation scheme 1, n = 50 1/h. The relative humidity in the ventilation air space in two climates during the four-year simulation period.

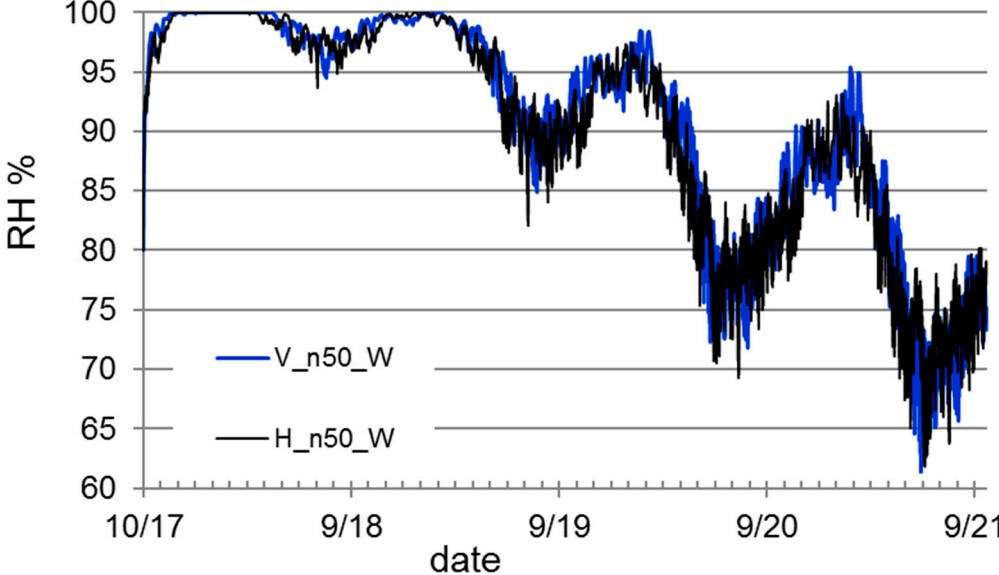

**Figure 20.** Concrete sandwich wall, wet exterior core, ventilation scheme 1, n = 50 1/h. The relative humidity in the boundary between mineral wool back layer insulation and the PU-insulation of the panel in two climates during the four-year simulation period.

In the case with ventilation rate n = 10 1/h, there was no condensation under the Madrid climate. In cold and mild climates, long periods of condensation conditions appeared inside the renovated structure system. Again, the condensation time did not depend much on the ventilation on the outer side of the PU-insulation. The condensation conditions lasted close to 5 months during the first year. Even if the condensation time is significantly shorter than in the brick wall case, the results refer to risks in the moisture performance with the studied ventilation. When the ventilation takes place only on the exterior parts of the renovation panel that is insulated from inside with relatively vapor tight PU-layer, the renovated structure system is not able to dry out from additional moisture in the structure without interstitial condensation conditions. The risks are higher under cold or mild European climate conditions.

This ventilation can be safely applied under these cold and mild climates only if the wall structure is dried out before the assembly of the renovation module to moisture contents below 80% RH level.

The use of a more vapor open thermal insulation product in the renovation panel could improve the drying efficiency of the system. However, with limited wall ventilation the condensation conditions would be at least partly transferred from the internal boundary to the outer parts of the system, which would likely cause different problems in the technical systems of the modules. This is due to the fact that the maximum ventilation rate in wall ventilation scheme 1 is quite limited.

### 4.4. Wall Ventilation Using Continuous Air Cavity between Material Layers

In ventilation scheme 2 the renovated brick wall has ventilation through a continuous air cavity that is between the thermal insulation layers: Back layer mineral wool insulation and the PU-insulation of the renovation module (Figure 7).

In this ventilation scheme (continuous air cavity), the ventilation air flow rate can be significantly higher than in the system with local ventilation channels (ventilation scheme 1). In addition, the higher temperature of the inner ventilation cavity enhances the drying. The ventilation affects the thermal performance of the wall more than in the case where the ventilation is close to exterior surface. The effect of the exterior thermal insulation on the thermal resistance of the wall can be significantly reduced due to the ventilation with outdoor air. The following results present the effect only on the moisture performance of the wall.

The critical part in the renovated structure is now the ventilation cavity between the PU and mineral wool insulations. The relative humidity of only this part are presented in the following. Figure 21 presents the results in the case with wall ventilation air flow rate n = 10 1/h in the 10 mm air cavity and Figure 22 in the case when n = 50 1/h.

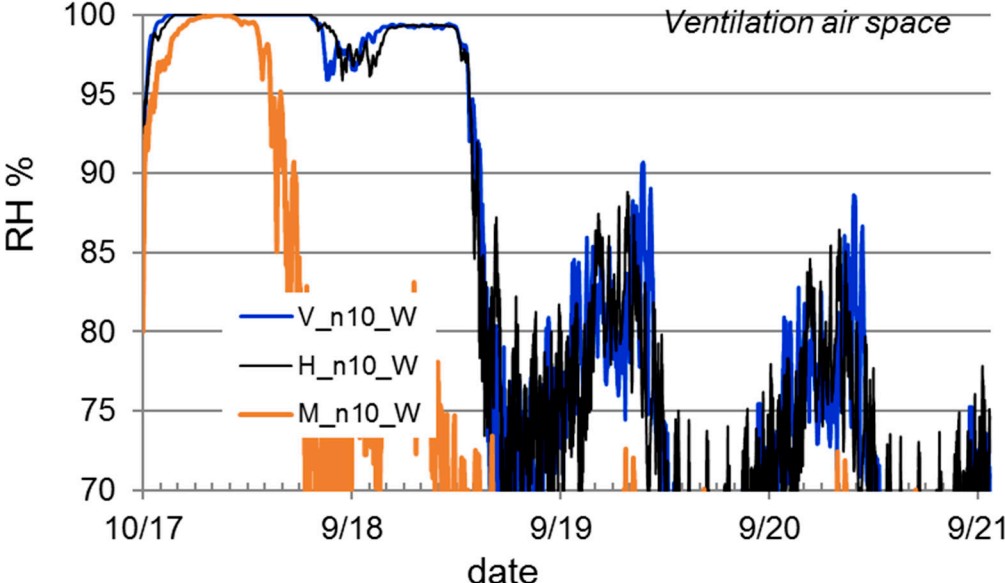

**Figure 21.** Brick wall, wet exterior core, ventilation scheme 2, n = 10 1/h. The relative humidity in the ventilation air space in the boundary between mineral wool back layer insulation and the PU-insulation of the panel in three different climates during the four-year simulation period.

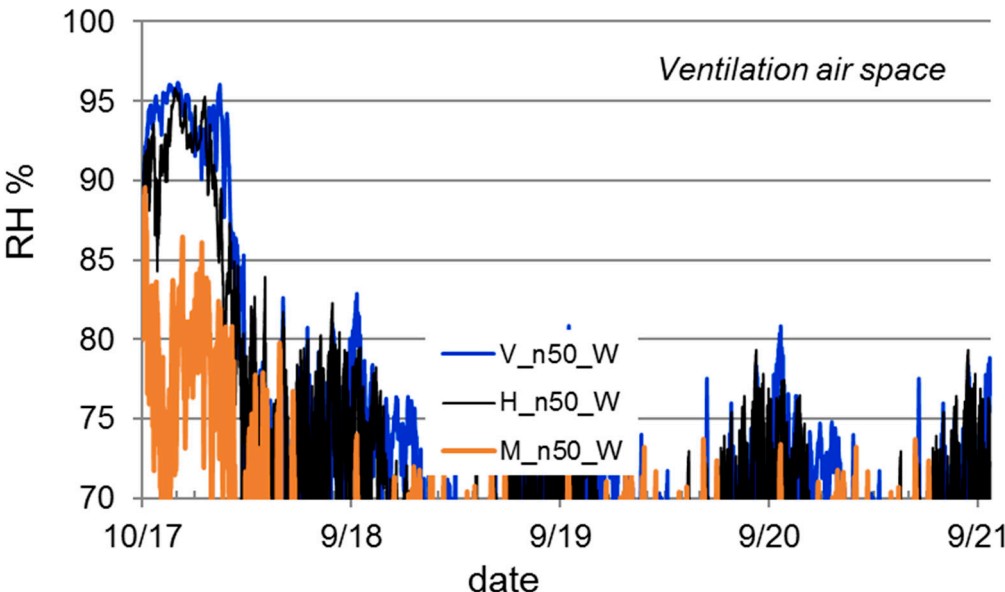

**Figure 22.** Brick wall, wet exterior core, ventilation scheme 2, n = 50 1/h. The relative humidity in the ventilation air space in the boundary between mineral wool back layer insulation and the PU-insulation of the panel in three different climates during the four-year simulation period.

In the case with n = 10 1/h ventilation rate, there were no condensation conditions in Madrid climate. In Vantaa, Finland and in Holzkirchen, Germany, there was about an 8-month condensation period during the first year when the additional moisture in the old wall was redistributed and dried out through the ventilation.

In the case with n = 50 1/h ventilation rate, there were no condensation conditions in any of the climates. High enough wall ventilation in this part of the wall makes it possible to dry out the initially high amounts of additional moisture without causing risks for condensation and moisture accumulation in the structure system.

It seems that when the old wall has high initial moisture content, the ventilation (when applying ventilation scheme 2) 10 1/h in a 10 mm continuous cavity is not enough to ensure safe moisture performance under cold and mild climate conditions. Ventilation rate 50 1/h seems to have some safety. The design of the ventilation system to reach the needed average ventilation flow rate and drying efficiency depends on the climate and initial moisture loads.

## 4.5. Evaluation of the Different Wall Ventilation Schemes

Two different wall ventilation schemes were studied (Figure 7). The moisture performance analysis shows that the ventilation scheme 1 (limited ventilation through local routes) can be safely applied only when the moisture content of the wall to be renovated is low enough (typically below 80% RH). When installed in warm climate conditions, the concrete core element may have higher initial moisture contents than the brick walls, but the performance should be checked before installation.

Ventilation scheme 2 can ensure safe moisture performance. The wall ventilation should be designed and set adequate for the moisture performance, because the ventilation always affects the thermal performance of the wall, and the possible benefits of the new thermal insulation can be easily lost with overly high ventilation rates.

It is obvious that the presented renovation modules can cause moisture performance risks for structures in any European climate if the drying of moisture to outdoors is prevented or left without consideration in the design.

*4.6. Effect of Ventilation on the Heat Losses*

The wall ventilation air flow rates were relatively low, but when the ventilation takes place inside the wall between thermal insulation layers (ventilation scheme 2), the cooling effect of the ventilation dilutes part of the effect of the thermal resistance of the system.

The n = 50 1/h air changes in a 10 mm air cavity corresponds to about 0.14 $dm^3$/s/$m^2$ air flow rate solved for the wall area. The low ventilation flow can be assumed to warm up in the in the ventilation cavity to the same level as what the boundary has in the pure heat conduction case. The ventilation represents additional heat losses. Assuming U-value of 0.17 W/$m^2$K for the renovated wall, the 50 mm exterior thermal insulation represents about 34% thermal resistance of the whole structure. With air change rate n = 50 1/h the ventilation heat losses are about 35 % of that of the nominal conductive heat losses of the wall. This means that the exterior 50 mm PU insulation brings practically no benefit to the thermal performance of the wall, even if the U-values solved under non-ventilated cases show high benefits.

## 5. Discussion and Conclusions

This paper focused on the performance aspects of the Meefs systems and especially on the building physics aspects of the renovation system. The prefabricated modular Meefs system is targeted at energy efficient façade renovation of typical apartment buildings. The façade design strategies and needs vary from climate to climate [71]. Therefore, a product designed for one set of climatic conditions may not be feasible in another climate. This paper addressed this challenge by analyzing thermal and moisture concerns raised during the development of the Meefs system. In addition, other properties needing careful design were assessed.

The hygrothermal performance of the renovation system was studied numerically and the results for 26 cases in three different climates were presented.

The following essential conclusions can be drawn from these analyses (majority of these are relevant also to any other comparable renovation solution):

- Any exterior renovation system should be able to dry out also outwards under different European climate conditions.
- The drying ability of the system should be ensured in the design phase of the systems.
- When the renovation panel forms a nearly vapor tight outer layer, drying can be achieved only by using ventilation of the structure.
- The possible ventilation rate and drying efficiency depends on the ventilation flow route in the renovated system.
- The drying of initially wet wall structures through the renovated system was possible only when the ventilation air flow rate was high enough (50 air changes in a 10 mm continuous one floor high air cavity) and the ventilation took place between the new insulated module and the back layer insulation. In this case, the effectiveness of the thermal insulation on the exterior side of the ventilation cavity will be reduced severely.
- When the ventilation takes place through local ventilation channels (edge area of the frames) or grooves in thermal insulation close to the exterior surface of the technical module, the drying efficiency remains relatively low. In this case, the existing structure should be dried out before the installation of the Meefs system to a moisture content corresponding to 80% RH equilibrium.
- The requirements for drying efficiency and the ventilation of structures are emphasized in cold and mild climate conditions. In some limited cases, the system can have safe performance under Madrid climate, but the risks are clear in Finnish and German climate conditions.
- Any exterior renovation system of the building façades should be designed from the beginning so that the total performance of the system is taken into account. This means that the drying efficiency, possibly needed ventilation scheme and the true effects on the thermal performance are studied and the system can be applied in practice without risk of malfunctions.

- Ventilation always affects the thermal performance of structures. With air change rate n = 50 1/h between the technical module and installation insulation, the heat losses practically nullify the effect of the thermal insulation on the exterior side of the ventilation cavity. Thus, the real thermal performance does not correspond to the U-values solved under non-ventilated conditions.

In practice, it is very difficult to adjust the wall ventilation to be just sufficient for the moisture performance needs. The effect of wind changes continuously, and it is more likely to have ventilation rates that highly exceed the need than to have controlled exact conditions. Thus, the additional heat losses though the ventilation may exceed the design values, which makes it even more challenging to reach the energy saving benefits promised for the renovation system.

When designing new renovation systems like the exterior panel system, one should always take into account the need for safe moisture performance. The system should be able to dry out the initial moisture of the old renovated façade without causing condensation and moisture accumulation in the system.

The Meefs system can offer significant benefits in energy efficiency, protection of old structures and integration of structures with other systems when applied in renovation of wall structures. This requires proper design of the total performance of the system including adequate ventilation within the structure.

**Author Contributions:** Conceptualization, S.P.; Methodology, S.P., T.O. and I.P.S.; Software, T.O.; Validation, T.O.; Formal Analysis, S.P. and T.O.; Investigation, S.P. and T.O.; Writing-Original Draft Preparation, S.P. and T.O.; Writing-Review & Editing, S.P., T.O., I.P.S. and M.P.; Visualization, S.P. and T.O.; Supervision, S.P., T.O. and I.P.S.; Project Administration, T.O. and I.P.S.; Funding Acquisition, I.P.S.

**Funding:** This project has received funding from the European Union's FP7 programme under grant agreement No 285411 (MeeFS Retrofitting. Multifunctional Energy Efficient Façade System for Building Retrofitting).

**Acknowledgments:** The results of this study are partly based on work done in the FP7 project "MeeFS Retrofitting. Multifunctional Energy Efficient Façade System for Building Retrofitting" (www.meefs-retrofitting.eu) co-financed by the European Commission and developed together with the following partners: Acciona Infrastructures, Tecnalia, AST ingenieria, Gobierno de Extremadura (Spain), E&L Architects, Greenovate Europe, CQFD Composites, TBC générateurs d'innovation, Antworks, Vipiemme solar, G.K. Rizakos, Ska Polska, National Technical University of Athens, Fraunhofer IAO, VTT Technical Research Centre of Finland and, Technion – Israel Institute of Technology.

**Conflicts of Interest:** The authors declare no conflict of interest.

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
