# Peer review of "Critical Performance Aspects of Retrofitting Apartment Buildings Using a Multifunctional Façade System"

_buildings, doi:10.3390/buildings9080184_

Round 1
Reviewer 1 Report
Review comments on “Critical performance aspects of retrofitting apartment buildings using a multifunctional façade system”.
The topic is not new, nevertheless, it is of some interest for the readers of Buildings.
The overall recommendation is to reconsider this article after a MAJOR revision.
My main comments are as below :
GENERAL COMMENTS:
- Authors (firstname + lastname) and affiliations are missing.
- Written English requires further editing. The manuscript suffers from typing and grammar errors and should be thoroughly read from an expert or English native speaker to ensure that your paper's text is polished and easy to read.
- Abbreviations: All of the abbreviations should be defined in full prior to it use. Please add a specific table with all abbreviations.
- References must be numbered in order of appearance in the text (including table captions and figure legends) and listed individually at the end of the manuscript. In the text, reference numbers should be placed in square brackets [ ] (not in apex format), and placed before the punctuation; for example [1], [1–3] or [1,3]. Please check all manuscript.
- The sections titles of the manuscript is not fully compliant with the instruction provided: please, re-edit it accordingly. Please see: https://www.mdpi.com/journal/buildings/instructions. There are no strict formatting requirements but the use of "Introduction", "Materials and Methods", "Results and Discussions" and "Conclusions" improves the paper readability.
- Line 21: Change "Introduction and literature review" in "Introduction".
- Line 22: I suggest to remove "e.g" before the reference numbers. Please check all the manuscript.
ABSTRACT
- The abstract should be rewritten in a better way. Needs to be explicitly updated with the main findings of the manuscript. Give in the last sentences of the abstract the main results of your work.
KEYWORDS:
- Do not repeat words of phrases from the title of the manuscript. Come up with new ones which will serve for indexing purposes of aims, objectives, methods and conclusions of the manuscript.
Line 148: Please improve the table 1 readability.
Line 149: Please improve the table 2 readability.
Line 190: Add x/y title and unit.
Line 357: I suggest to Change "1" in "4.1.1".
Line 366: I suggest to Change "2" in "4.1.2".
Line 429: Add x/y title.
Line 696-710: Author Contributions, Funding, possible acknowledgments/Conflicts of Interest are missing.
REFERENCES
- The reference list at the end of the manuscript is not fully compliant with the instruction provided: please, re-edit it accordingly.
Please see: https://www.mdpi.com/journal/buildings/instructions#references
Reviewer 2 Report
The study is well done and articulated; the topic is of great interest and is studied critically and comparatively.
I suggest some evaluations:
- Reference 1 is a book or an article ? Is not clear.
- in the introduction you could mention not only the technological obsolescence but also the social obsolescence, the latter caused by the new needs of family groups for residential buildings
- Line 112 “façade concept based on multi-module technology components ..” the system is only modular or is possible talking about its flexibility ?
- Please check the quality of figure 1
-Regarding apartments in Germany, does the study also consider those in the ex East Germany? Are there differences with those of the ex West Germany?
- in point 3.3, perhaps a figure (technical detail) would be appropriate
Reviewer 3 Report
Lines 22–23: Instead of referencing articles with e.g. [#], author[s] should use (author, date) method in text.
Line 85: Change "which" to "that".
Lines 285–286: Revise to read "A vapor barrier that is too tight can cause moisture accumulation .."
Round 2
Reviewer 1 Report
I would like to thank the authors for their efforts to improve their manuscript.
Further comments here below:
- Please check all "Error! Reference source not found" in the manuscript (e.g. line 144; line 207; ...)
- The reference list at the end of the manuscript is not fully compliant with the instruction provided! Please, re-edit it accordingly. Please see: https://www.mdpi.com/journal/buildings/instructions#references
Author Response
References corrected.
” - Please check all "Error! Reference source not found" in the manuscript (e.g. line 144; line 207; ...)”. Is not visible in our Word version.